# Malingering and Stimulant Medications Abuse, Misuse and Diversion

**DOI:** 10.3390/brainsci12081004

**Published:** 2022-07-28

**Authors:** Joseph Sadek

**Affiliations:** Department of Psychiatry, Dalhousie University, Halifax, NS B3H 2E2, Canada; joseph.sadek@nshealth.ca

**Keywords:** ADHD, malingering, diversion, stimulant medications abuse, misuse

## Abstract

Attention deficit hyperactivity disorder (ADHD) is a neurodevelopmental disorder that interferes with multiple aspects of daily functioning. Malingering or feigning of symptoms can be a major challenge during ADHD assessment. Stimulant medication abuse, misuse and diversion may constitute another challenge during management. A literature search of the past 15 years on the topic continued to suggest that there are several reasons for malingering and faking ADHD symptoms. Some of the reasons include the intent to obtain prescriptions for stimulant medications for performance enhancement, to gain access to additional school services and accommodations, to use recreationally and to sell as a street drug. In some countries, patients may receive additional tax or student loan benefits. Several researchers suggested that self-report rating measures are easily simulated by patients without ADHD. They concluded that no questionnaire has proved sufficiently robust against false positives. Some clinical factors that may suggest malingering during the ADHD assessment are highlighted and some available tests to detect malingering are discussed.

## 1. Introduction:

The concept of feigning of mental illness is not new. The feigning of mental illness for external incentive has been recognized for centuries. In 1825, some authors stated: “Diseases are generally feigned for one of three causes—fear, shame, or hope of gain.” They also noted that the most easily feigned illnesses are those with few to no physical manifestations or those based on self-report and, in particular, “insanity, epilepsy, and pain”. They suggested five elements to consider malingering: “An external incentive is present, no causative factor is present, or the illness has a sudden onset, the individual is resistant to receiving treatment, symptom complaints are inconsistent with the true illness and the course of the disorder is inconsistent with the true illness” [1].

Attention deficit hyperactivity disorder (ADHD) is a neurodevelopmental disorder that interferes with multiple aspects of daily functioning. There is a need for a clear and concise approach to the diagnosis of a complex disorder such as ADHD. The diagnosis and management of ADHD in adults is complex and challenging for many reasons. One reason is the presence of comorbidity of other psychiatric disorders that have symptoms overlapping with those of ADHD. The presence of comorbidities can be challenging in terms of both diagnosis and management. Another challenge in ADHD assessment is malingering and feigning of symptoms. Clinicians rely on the self-report of ADHD symptoms in addition to collateral information, but the subjective report may be inaccurate or misleading. Understanding the diagnosis and management of ADHD is incomplete without understanding the full dimensions of malingering and feigning of symptoms [2].

Misuse or nonmedical medication use (NMU) can include abuse, inappropriate use and dependence, as was described in the DSM-IV and replaced by stimulant use disorder in the DSM-5 [3]. Medication misuse occurs when the medication is used for purposes that are different from those that were prescribed, including performance enhancement and recreational use. Medications associated with misuse and abuse are often diverted (i.e., sold into an illegal market for high monetary value and used in a way that is generally not medically authorized). Financial gains from selling medications may encourage malingering [4].

The DSM-5 defines malingering as “the intentional production of false or grossly exaggerated physical or psychological symptoms, motivated by external incentives” [3]. Prescription stimulants, specifically methylphenidates (MPHs) and dextroamphetamine-amphetamines (AMPs), are classified as controlled substances and commonly prescribed for pharmacological treatment of ADHD. They are effective at managing symptoms of ADHD, but they also have abuse potential [4].

There are several reasons for malingering and faking ADHD symptoms. Some of the reasons include the intent to obtain prescriptions for stimulant medications for performance enhancement. Students may gain access to additional school services and accommodations (e.g., separate testing environments, quiet testing environments, extended testing time, reduced homework, provision of a note taker or even alternative courses). Some individuals obtain stimulants to use recreationally and to sell as a street drug. In some countries, patients may receive additional tax or student loan reduction benefits [4,5,6,7,8].

Given the large body of literature and space constraints, this review is selective rather than systematic and comprehensive. It relies on other systematic reviews and meta-analysis of the past 15 years in addition to clinical data and expert opinion. The article attempts to provide guidance on several questions that are relevant to clinicians such as the prevalence of malingering, reasons for malingering, what clinical factors that may suggest malingering during the ADHD assessment should be considered and what is the current evidence of tests used to detect malingering.

## 2. Summary of Some Previously Published Studies

### 2.1. Epidemiology

Some researchers suggested that life-time prevalence rates of nonmedical use of prescribed stimulant use (NMUPS) in college and university students ranges from 5% to 40%, but the exact prevalence is unknown [9,10]. They suggested that as many as 50% of students who self-referred for an ADHD evaluation on a US campus were thought to have exaggerated their symptoms. Their performance on neuropsychological assessments was suggestive of malingering [6]. Similar findings were reported by other authors [11]. Research on the feigning of ADHD outside of college populations is sparce; however, in one large study of adults in the US, nearly 20% of past-year nonmedical users indicated that they had obtained their medication fraudulently from a physician by misrepresenting their symptoms [12]. Another study found that 10.5% of female college students endorsed having ever used an ADHD-specific stimulant outside a physician’s prescription for the purpose of weight loss [13]. One study suggested that lifetime rates of stimulant diversion among college students ranges from 15% to 30% [14].

### 2.2. Strategies for Faking ADHD Diagnosis in the Selected Literature

Authors examined strategies used to fake ADHD diagnosis and concluded that specific strategies have been used to fake ADHD symptoms. They suggested that adult ADHD can be successfully faked on self-report instruments [15]. ADHD simulators try to show difficulty paying attention and attempt to appear less intelligent. They complete tasks slowly by trying to act like an acquaintance with ADHD. They also zone out or attend to distracting noises, choose incorrect answers, particularly on harder items, skip items, respond quickly and carelessly while completing tasks, appear inattentive to oral instructions or refuse to comply with the instructions, select items on the scale that matched DSM-5 criteria and only focus their eyes on the middle of the page. They begin tasks before being asked to begin, pretend to have forgotten things and act confused or nervous [15,16]. Some individuals feigning ADHD tend to endorse more symptoms of hyperactivity/restlessness compared with clinical ADHD groups, but these differences do not detect feigned ADHD [8,17].

### 2.3. ADHD Evaluation Tools and Malingering

The literature indicates that rating scales and questionnaires that are based on self-report are not sensitive enough to allow the detection of malingered ADHD symptoms [16,18,19]. Researchers suggested that self-report rating scales and questionnaires are easily simulated by individuals who do not have ADHD symptoms. They concluded that questionnaires and rating scales that are based on self-report may produce high false-positive results [2,10].

Several researchers who examined simulation studies on personality inventories such as the Minnesota Multiphasic Personality Inventory, Second Edition (MMPI-2) and its Restructured Form (RF) concluded that they are more useful than self-report ADHD questionnaires in detecting simulated ADHD, but more research is needed [2,8,20]. The F (Infrequency) scale in MMPI-2 is used to detect attempts of faking good or faking bad. A high F score indicates that the individuals taking the test are trying to appear better or worse than they really are. There is a specific validity index embedded in the MMPI-2 and MMPI-2 RF that can be particularly helpful in detecting simulated ADHD. The index is designed to measure infrequently endorsed items related to psychopathology and labeled as Fp-r (Infrequent Psychopathology Responses) [2].

Several neuropsychological tests have been used during the diagnostic assessment of ADHD: Test of Variables of Attention (TOVA) [21], Integrated Visual and Auditory Continuous Performance Test (IVA CPT) [22], Woodcock–Johnson Psychoeducational Battery [23], Conners’ Continuous Performance Test-II (C-CPT-II) [24], Trail Making Test Parts A and B [25], and Neuropsychological Assessment Battery Numbers and Letters Test Part A (NABNLA) [26]. The literature suggests that neuropsychological test data alone are insufficient for detecting feigned ADHD and that individuals who feign ADHD symptoms can produce cognitive profiles that resemble patients with ADHD. The validity indexes embedded in some of these batteries show some promise in detecting malingering [11,19,22,27,28,29].

The term performance validity refers to validity of test performance. A performance validity test (PVT) such as the Test of Memory Malingering (TOMM) may be used to detect malingering during cognitive assessment. A performance validity test can be a separate test or part of another scale. It has high specificity when the traditional cut-off points are used to identify irregular performance in cognitive tests [30].

Symptom validity testing (SVT) can be a separate scale or embedded in neuropsychological test batteries and used to test the validity of a symptom report. There are two general categories of SVTs: cognitive and psychological. Examples of SVTs include the Recognition Memory Test (RMT), Word Memory Test (WMT), Portland Digit Recognition Test (PDRT), Victoria Symptom Validity Test (VSVT), which originated in Canada, and Validity Indicator Profile (VIP) [31].

It appears that SVTs can be useful in detecting malingered ADHD, but was originally designed to detect malingered cognitive symptoms. SVTs have a high specificity, but inadequate sensitivity. Combining two SVTs may significantly increase the ability to identify malingerers [30]. A recent meta-analysis suggested that, overall, PVTs (stand-alone and embedded) produced a large effect size, whereas overall SVTs (stand-alone and embedded) produced a medium-effect size. The meta-analysis suggested that performance validity tests are more effective than symptom validity tests in detecting college students attempting to feign ADHD diagnosis [32].

Clinicians are encouraged to consider the following clinical factors that may suggest malingering during the ADHD assessment:Marked disagreement and contradictions of the reported symptoms. Example: patient reported inability to complete tasks because of inattention, but the patient also reported finishing all the required tasks such as completing assignments on time.Marked disagreement and discrepancies between reported and observed symptoms. Example: patient reported blurting answers before the questions are complete, but during a 2 h assessment, no blurting of answers observed.Marked discrepancies between reported and actual level of function. Example: reported poor academic functioning in university, but the records show a GPA average of 4.1 each year.Lack of evidence or discrepancies between reported symptoms and psychological test results. Example: patient mentioned that he was diagnosed with ADHD in grade 5, but his family physician included in the referral letter the results of his psychoeducational assessment that shows that patient was not diagnosed with ADHD.Patient endorses improbable ADHD symptoms. Examples include forgetting the names of siblings, directions to go home and name of elementary school.Patient is behaving in an unusual way during the interview. Examples include extremely guarded, uncooperative, angry, evasive and responding with difficulty answering simple questions.

This opinion is limited by the clinical experience in Western society and may apply differently in other countries.

## 3. Conclusions

This article aims to underscore the importance of the problem of malingering and stimulant use disorder. Malingering for ADHD medications is difficult to detect. Stimulant medication abuse, misuse and diversion constitute a serious public health concern. These medications have major adverse effects and addictive potential. It is suggested that malingering is not an infrequent event and has been increasing overtime [2,6,11]. Public health interventions such as increasing public and clinical awareness of risks related to NMUPS should be encouraged. Modifying treatment guidelines and establishing quality indicators that assess treatment adherence would be a helpful step in patient care. Clinicians should be encouraged to consider the possibility of malingering and add it to their differential diagnosis if suspected. Performance-based measures should be included in ADHD evaluation batteries. ADHD assessment should include interviewing of a parent or significant other, obtaining collateral information from different sources, reviewing past educational records, medications, medical records, obtaining all past psychiatric/psychological records, asking about trouble with the law and any forensic history and clarifying secondary gains. Clinicians are encouraged not to be rushed into filling out disability forms nor prescribing stimulant medication when the diagnosis is unclear. Spending more time completing a full ADHD assessment would be more valuable than reaching an inaccurate conclusion. In the future, there might be a potential use of the objective measures of ADHD based on neuroimaging methods to avoid the feigning of symptoms [33,34]. More research is needed in addressing ADHD and malingering.

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
