# Peer review of "Malingering and Stimulant Medications Abuse, Misuse and Diversion"

_brainsci, 2022, doi:10.3390/brainsci12081004_

Round 1

Reviewer 1 Report

Major

1.      The second paragraph of the introduction has to focus more on the fringing of the ADHD symptoms rather than describing the diagnosis challenges. A better link it to the first paragraph is lacking.    

2.      I encourage author to briefly discuss the objective measures of ADHD based on neuroimaging methods and the potential use of them to avoid fringing of symptoms. Please see:

·        Kaboodvand, N., Iravani, B., & Fransson, P. (2020). Dynamic synergetic configurations of resting-state networks in ADHD. Neuroimage207, 116347.

·        Iravani, B., Arshamian, A., Fransson, P., & Kaboodvand, N. (2021). Whole-brain modelling of resting state fMRI differentiates ADHD subtypes and facilitates stratified neuro-stimulation therapy. Neuroimage, 231, 117844.

3.      I encourage author to discuss the limitation of their work. For example, the review seems to cover only western societies and to a large extent North America. This has to be openly discussed as limitations.

Minor

4.      DSM sometimes written in roman number (DSM-V) and sometimes not (DSM-5). Please keep it consistent across the paper.

Author Response

Dear Reviewer

       Thanks so much for you excellent and helpful comments. I incorporated all the suggestions into the manuscript. I appreciate your expertise and input in this important area.

       Thanks again

       Joe Sadek

Reviewer 2 Report

The opinion by Ass. Professor J Sadek is discussing potential issue surrounding malingering and faking ADHD symptoms to gain prescription of psychostimulants. In this manuscript, the author also discusses abuse, misuse and diversion of psychostimulants, mainly MPH (amphetamines are also mentioned).

The article is easy to read and follow. It presents a very nice insight of the author’s opinion of ADHD and stimulants. However, a few corrections are needed before publication. These are detailed line-by-line below, and the author is advised to refer to the enclosed PDF document for yellow highlighting.

-General comment, since this is not a comprehensive appraisal of the literature, it might be needed to not use the standard layout of a research article (with intro, methods, results and discussion). For example, the section “3. Results” is just a summary of some previously-published studies, and thus, are not new “results”.

-General comment, please improve the grammar.

-Abstract, line 6 and Introduction line 31. Missing word.

-Introduction, line 22, grammar.

-Introduction, lines 27, 32, 33, 34, 94, 114, 115, 118, 121, 135, odd spacing.

-Introduction, line 34, missing word “for” (challenging for many reasons).

-Introduction, lines 38-40, incorrect sentence. The fact that stimulant medication can be diverted, misused and abused is totally irrelevant to the diagnosis and management of ADHD, as some patients do need medication. This needs rephrasing.

 -Results, line 52, missing “the” (in the DSM-4 [...] in the DSM-5).

-Results, line 57, remove the comma.

-Results, line 61, duplication of “controlled” substances. Please rephrase. Does the authors mean something along the line of “controlled substances classified as Rx-only medications” ?

-Results, lines 63, 70, 107, 118, abnormal punctuation.

-Results, line 81, abnormal spacing.

-Results, lines 83 and 93, abnormal use of a capital letter.

-Results, line 108, please describe in full the first term (“Fp”).

-Results, line 115, a reference is found without brackets.

-Results, line 119, please confirm that “determination” is the adequate terminology here.

-Results, lines 151-154. Here, the author here is describing a very interesting fact, lack of evidence. Thus, please add “lack of evidence” in this paragraph.

-Conclusion, line 176. Missing word. “Would be more valuable” ?

-References, line 183. This book by Beck & Dunlop was published in 1825. It appears that “Anderson et alia” is the publisher/printer. Please also add the printing location: London.

Author Response

       Dear Reviewer

       Thanks so much for you excellent and helpful comments. I incorporated all the suggestions into the manuscript. I appreciate your expertise and input in this important area. I am very grateful for all your help and support in reviewing this manuscript.

       Thanks again

         Joe Sadek

-General comment, since this is not a comprehensive appraisal of the literature, it might be needed to not use the standard layout of a research article (with intro, methods, results and discussion). For example, the section “3. Results” is just a summary of some previously-published studies, and thus, are not new “results”.

Results changed to a summary of some previously-published

-Abstract, line 6 and Introduction line 31. Missing word. corrected

-Introduction, line 22, grammar. corrected

-Introduction, lines 27, 32, 33, 34, 94, 114, 115, 118, 121, 135, odd spacing. Corrected

-Introduction, line 34, missing word “for” (challenging for many reasons). Corrected

-Introduction, lines 38-40, incorrect sentence. The fact that stimulant medication can be diverted, misused and abused is totally irrelevant to the diagnosis and management of ADHD, as some patients do need medication. This needs rephrasing .corrected..

 -Results, line 52, missing “the” (in the DSM-4 [...] in the DSM-5). corrected

-Results, line 57, remove the comma. corrected

-Results, line 61, duplication of “controlled” substances. Please rephrase. Does the authors mean something along the line of “controlled substances classified as Rx-only medications” ? Duplication removed.

-Results, lines 63, 70, 107, 118, abnormal punctuation.

-Results, line 81, abnormal spacing. corrected

-Results, lines 83 and 93, abnormal use of a capital letter. corrected

-Results, line 108, please describe in full the first term (“Fp”). Corrected Infrequent psychopathology

-Results, line 115, a reference is found without brackets. Corrected ref 25

-Results, line 119, please confirm that “determination” is the adequate terminology here. Corrected multiple sources for determination of malingering during ADHD assessment [19].

-Results, lines 151-154. Here, the author here is describing a very interesting fact, lack of evidence. Thus, please add “lack of evidence” in this paragraph. Corrected. Added

-Conclusion, line 176. Missing word. “Would be more valuable” ? corrected

-References, line 183. This book by Beck & Dunlop was published in 1825. It appears that “Anderson et alia” is the publisher/printer. Please also add the printing location: London. corrected

Reviewer 3 Report

This is a very important opinion,

on a very important topic. I have no comments,

as I agree.

Please just fill the blank spaces:

Author Contributions: 178 Funding: 179 Acknowledgments: 180 Conflicts of Interest:

Author Response

Dear Reviewer

Thanks for much for the time, effort and expertise you put into this review. 

I am very grateful.

Thanks

Joe Sadek 

Round 2

Reviewer 1 Report

I would like to thank the author for fully answering my concern. I have no further comment and recommend this work for publication.